

# Depression and low physical activity are related to sarcopenia in hemodialysis: a single-center study

Kornanong Yuenyongchaiwat[1,2], Sasikan Jongritthiporn[1], Kasarn Somsamarn[1], Oranat Sukkho[1], Sasipim Pairojkittrakul[3] and Opas Traitanon[4]

[1] Department of Physiotherapy/ Faculty of Allied Health Sciences, Thammasat University, Pathumthani, Thailand
[2] Thammasat University, Research Unit in Physical Therapy in Respiratory and Cardiovascular Systems, Pathumthani, Pathumthani, Thailand
[3] Nephrology Clinic, Thammasat University Hospital, Pathumthani, Thailand
[4] Division of Nephrology, Department of Internal medicine, Faculty of Medicine, Thammasat University, Pathumthani, Thailand

Corresponding author
Kornanong Yuenyongchaiwat,
ykornano@tu.ac.th

## ABSTRACT

**Background**. The number of patients who suffer from chronic renal failure (CRF) has widely increased worldwide. Patients with advanced stages of CRF experience a gradual and progressive loss of muscle and fat mass leading to decreased physical activity and mental health problems. The loss of muscle mass in CRF might contribute to the development of sarcopenia. Therefore, this study aimed to explore the prevalence of sarcopenia and to determine the relationship of physical activity and mental state of depression with sarcopenia in hemodialysis patients.

**Methods**. A cross-sectional study was designed with a total of 104 male and female with a minimum age of 35 years. Based on the guidelines of the Asian Working Group for Sarcopenia in 2019, gait speed, muscle mass, and handgrip were used to define sarcopenia. In addition, participants were requested to perform a set of questionnaires to evaluate their physical activity and state of depression. Logistic regression analyses were used to explore the risk factors of sarcopenia.

**Results**. Thirty-four (32.69%) of 104 participants had sarcopenia. Compared to the 70 individuals without sarcopenia, they had a low physical activity and a high depression score (ps < .05). Furthermore, low physical activity and high depression scores in combination with sarcopenia were associated with an increased mortality risk. Low physical activity and high depression scores were also independently associated with sarcopenia in hemodialysis patients after controlling for age (odds ratio = 3.23, and 4.92, respectively).

## INTRODUCTION

Chronic renal failure (CRF) is a common worldwide health problem including in Thailand. A 17.5% prevalence of CRF was reported in Thailand with a prevalence of 7.8% in males and 9.3% in females at stages 3–5 (*Ingsathit et al., 2010*). Prevalence of CRF stages 4–5 was increased to 3.6%, 7.2% and 12.7% at the age of 60, 70, and over 80 years, respectively

(*Krittayaphong et al., 2017*). Also, over 100,000 of the CRF patients in Thailand underwent hemodialysis (*Kanjanabuch & Takkavatakarn, 2020*). The comorbidity of hemodialysis patients is associated with physical health dysfunction, poor physical activity, and mental health problems. These will lead to a poor quality of life. Regarding depression, it has been reported that mental health complications, in particular higher depressive symptoms, are commonly observed in CRF patients. Furthermore, a systematic review and metanalysis revealed that the relationship between lower albumin in CRF and end stage renal failure is associated with severe depressive symptoms (ascertained by self-reported measures) (*Gregg et al., 2020*). Therefore, protein-energy wasting was proposed as a potential mechanism underlying the high comorbidity of depression. In addition, hemodialysis patients had a low level of physical activity and more sedentary lifestyle compared with age- and gender-matched individuals without CRF (*Gomes et al., 2015*). Uremic myopathy that leads to muscle wasting and poor muscle performance has been reported in patients with CRF undergoing hemodialysis (*Fahal, 2014*). Recently, sarcopenia has been defined in terms of loss of muscle mass and it is categorized into primary and secondary sarcopenia (*Cruz-Jentoft et al., 2019*). Muscle atrophy, less activity and also reduced physical function has been observed in dialysis patients and might be caused by uremic myopathy (*Fahal, 2014*). Therefore, the purpose of this study was to explore the prevalence of sarcopenia and the relationships with physical activity and depressive symptoms in Thai CRF patients undergoing hemodialysis.

## MATERIALS & METHODS

This study was carried out according to the principles of the Declaration of Helsinki 1975. The Research Ethics Committee on Human Subject of Thammasat University (approval reference COA No. 117/2562) and the Ethics in Human Research Committee of the Thammasat University Hospital approved the study. Information sheets and the consent form were given to the participating patients prior to the study. The Clinical Trials Registry (TCTR) is TCTR20190911004. The following formula was used to calculate the sample size, $n = N / [1+Ne^2]$; where N = total number of hemodialysis patients at the Hemodialysis Center of Thammasat University Hospital, n = number of samples, and e = error tolerance = 0.05. According to a report from the Hemodialysis Center of Thammasat University Hospital in 2018, the number of hemodialysis patients was 140. Therefore, a total of 104 Thai patients who underwent hemodialysis were recruited. The minimum age of the male and female participants was 35 years. These patients required hemodialysis at least two times per week. According to American College of Cardiology and American Heart Association recommendations, resting blood pressure >180/120 mmHg is defined as hypertensive and should be see a doctor (*Whelton et al., 2018*). In addition, postural orthostatic tachycardia syndrome is assessed by resting heart rate ≥120 beats per min (*Schmidt, Karabin & Malone, 2017*). Therefore, these conditions were excluded in the present study. In addition, participants who had a high fever within 24 h prior to the test, psychiatric problems or pregnant women were also excluded. Participants were asked to complete a set of questionnaires, i.e., Beck Depression Inventory-II (BDI-II) and

Global Physical Activity Questionnaire (GPAQ). BDI-II comprises 21 items with higher scores indicating greater depression. A total score of 13 or less is interpreted as minimal depression. A score between 14 and 19 is interpreted as mild depression, a score of 20–28 is interpreted as moderate depression, and a score of 29–63 is interpreted as severe depression (*Beck, Steer & Brown, 1996*). In addition, the GPAQ was developed by the World Health Organization and it has been commonly used to define physical activity level (*Armstrong & Bull, 2006*). A high physical activity is categorized as ≥1,500 MET minutes per week and low physical activity as is defined as <600 MET minutes per week (*Armstrong & Bull, 2006*). A screening sarcopenia in the present study followed a new version of Asian Working Group for Sarcopenia (AWGS) in 2019 which are poor muscle strength or slow gait speed plus loss of muscle mass (*Chen et al., 2020*). The applied AWGS 2019 cutoffs for sarcopenia diagnosis were as follows: slow gait speed (<1.0 m/s) or poor physical performance (handgrip strength <28 kg for men and <18 kg for women) plus low muscle mass ($<7.0$ kg/m$^2$ in men and $<5.7$ kg/m$^2$ in women). Therefore, all participants were requested to perform a 6-meter walking test (i.e., gait speed) and a handgrip strength test. A walking speed lower than 1.0 m/s was categorized as slow gait speed. In addition, poor muscle strength was categorized as handgrip strength less than 28 kg in men and less than 18 kg in women. Participants who had low handgrip strength and/or slow gait speed were required to evaluate the skeletal muscle mass (SMM). The participants were required to perform the bioelectrical impedance analysis (BIA: Omron HBF-375 body composition monitor; Omron Healthcare Co., Ltd., Japan) with body weight, BMI, and SMM also recorded. Skeletal muscle index (SMI) was calculated by SMM in kg adjusted for the squared height (SMM/hight$^2$, kg/m$^2$). A SMI value of $<7.0$ kg/m$^2$ in men and $<5.7$ kg/m$^2$ in women was defined as low muscle mass.

To compare between sarcopenic and non-sarcopenic patients with hemodialysis, $T$-test and chi-square test were used. Pearson correlation was used to determine the relationships of sarcopenia. Finally, logistic analysis was performed to explore the independent risk for sarcopenia, a $p$-value of 0.05 was considered statistically significant. IBM SPSS version 24.0 was used for statistical analyses.

## RESULTS

A total of 104 hemodialysis patients, 54 males and 50 females, were recruited and sarcopenia was found in 34 (32.69%) of them. Classification was due to either poor grip strength ($n = 4$, 3.85%), low gait speed ($n = 7$, 6.73%), or concomitant declined muscle strength and low gait speed ($n = 23$, 22.12%) Fig. 1.

The average age of the 104 participants was 59.74(13.62) years. The 34 participants with sarcopenia were observed in advanced age, lower BMI, and longer duration of hemodialysis compared to those without sarcopenia (Table 1). Regarding underlying disease, participants with sarcopenia were significantly affected by a history of diabetes mellitus (DM) ($\chi^2 = 5.861$, $P = 0.018$). Lower physical activity, higher depression scores and poor cognitive performance were also reported in the sarcopenia group (Table 1).

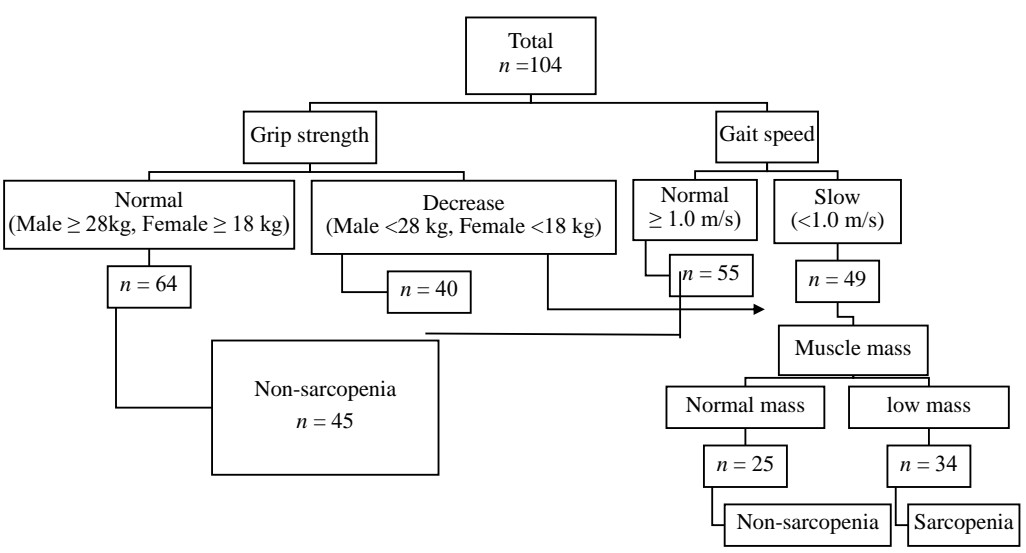

**Figure 1** **Diagnostic algorithms.** Diagnostic algorithms of Asian Working Group for Sarcopenia in chronic kidney disease.

The correlation of risk factors with sarcopenia was examined and the results are presented in Table 2. Risk factors, e.g., physical activity and depression were associated with sarcopenia in hemodialysis patients (Ps >.05).

To determine the risk factors for developing sarcopenia, a logistic regression analysis was used. Age, sex, BMI, history of DM, duration of hemodialysis, symptoms of depression and levels of physical activity were explored (Table 3). Depression was found to be a major risk factor (odds ratio (OR) 3.229, 95% confidence interval (CI) 1.139–9.157, p = .028) followed by history of DM (OR = .356, 95% CI [0.152–0.832], p = .017), duration of hemodialysis (longer than 10 years) (OR = 0.316, 95% CI [0.111–0.896], p = .030) and low physical activity (OR = 0.284, 95% CI [0.121–0.668], p = .004).

## DISCUSSION

The present study aimed to explore the prevalence of sarcopenia in CRF patients with hemodialysis and to also explore the relationships between sarcopenia, depression, and physical activity. Data from a total of 104 patients were analyzed and sarcopenia was detected in 34 patients (32.69%). This prevalence is high compared to the reported 16% overall prevalence of sarcopenia in elderly from Thailand (*Yuenyongchaiwat & Boonsinsukh, 2020*). The results show that secondary sarcopenia has a higher prevalence rate in CRF patients than primary sarcopenia has in the older adults in Thailand.

Advanced age, long duration of hemodialysis, low physical activity and symptoms of depression were also found associated with sarcopenia in hemodialysis patients. *Bataille et al. (2017)* found that 31.5% of hemodialysis patients was defined as sarcopenia and these were older, had longer dialysis, and low BMI. Regarding the European Working Group on Sarcopenia in Older People (EWGSOP) criteria, it was noted that the prevalence

**Table 1  Characteristic data of the participating hemodialysis patients with statistical analysis of differences between sarcopenic and non-sarcopenic patients.**

| | Total (n = 104) Mean(SD) | Sarcopenia (n = 34) Mean(SD) | No sarcopenia (n = 70) Mean(SD) | t (102) | P-value | 95% CI |
|---|---|---|---|---|---|---|
| Sex | | | | 0.021[*] | 1.000 | 0.414 to 2.138 |
|   Male (%) | 54 (51.92) | 18 (17.31) | 36 (34.62) | | | |
|   Female (%) | 50 (48.08) | 16 (15.38) | 34 (32.69) | | | |
| Age (years) | $59.74 \pm 13.62$ | $68.65 \pm 13.89$ | $55.41 \pm 11.25$ | 5.20 | <.001 | 8.188 to 18.277 |
| Underlying diseases | | | | | | |
|   Hypertension (%) | 74 (71.15) | 22 (21.15) | 52 (50.00) | 1.023[*] | .359 | 0.262 to 1.537 |
|   Diabetes mellitus (%) | 38 (37.54) | 18 (17.31) | 20 (19.23) | 5.861[*] | .018 | 1.202 to 6.580 |
|   Dyslipidemia (%) | 24 (23.08) | 7 (6.73) | 17 (16.35) | 0.176[*] | .806 | 0.299 to 2.186 |
|   Gout (%) | 11 (10.58) | 4 (3.85) | 7 (6.73) | 0.075[*] | .747 | 0.326 to 4.417 |
|   Heart disease (%) | 23 (22.12) | 9 (8.65) | 14 (13.46) | 0.556[*] | .461 | 0.551 to 3.765 |
| BMI (kg/m$^2$) | 23.48(4.11) | 21.11(2.47) | 24.64(4.26) | −4.471 | <.001 | −5.094 to −1.963 |
| Duration of hemodialysis (yrs.) | 5.86(4.95) | 7.41(5.70) | 5.10(4.39) | 2.280 | .025 | 0.301 to 4.325 |
| Gait speed (m/s) | 0.96(0.35) | 0.73(0.35 | 1.07(0.29) | −5.238 | <.001 | −0.468 to −0.211 |
| Grip strength (kg) | 23.30(8.41) | 17.26(5.63) | 26.23(7.98) | −5.878 | <.001 | −11.997 to −5.943 |
| Skeletal mass index (kg/m$^2$) | 6.21(1.29) | 5.18(0.95) | 6.71(1.12) | −6.846 | <.001 | −1.975 to −1.088 |
| Depression scores | 7.43(6.35) | 10.29(7.19) | 6.04(5.43) | 3.359 | .001 | 1.741 to 6.761 |
| Physical activity (MET minutes per week) | 1882.31(2946.57) | 592.06(662.66) | 2535.73(3383.55) | −3.408 | .001 | −3161.93 to −835.406 |

**Notes.**
[*]Chi-square test.
  BMI,  body mass index..

of sarcopenia in patients undergoing hemodialysis was 37% (*Kim et al., 2014*). A 33.3% prevalence of sarcopenia was reported in persons older than 60 years (*Ren et al., 2016*). *Mori et al. (2019)* reported a 40% prevalence of sarcopenia in patients undergoing hemodialysis. Here, the present study found the prevalence of sarcopenia was 32.69% by AWGS definition. Therefore, the reported prevalence of sarcopenia in hemodialysis patients varies somewhat, either due to definition (i.e., the European Working Group on Sarcopenia in Older People; EWGSOP, AWGS), measurement of muscle mass (e.g., DEXA, BIA), or method and cutoff. In all the hemodialysis patients had a prevalence rate of sarcopenia in the range of 31.5–40% (*Bataille et al., 2017*; *Kim et al., 2014*; *Mori et al., 2019*; *Ren et al., 2016*). Furthermore, longer duration of hemodialysis was closely associated with higher sarcopenia occurrence in the patients (*Ren et al., 2016*). CRF in hemodialysis patients has been found together with poor physical function due to loss of muscle strength caused by muscle atrophy and hypoplasia (*Fahal, 2014*). The mechanism of uremic change might contribute to the decrease in the synthesis of muscle protein and result in uremic sarcopenia (*Fahal, 2014*).

Regarding the comorbidities, the present study found that a history of DM was associated with sarcopenia in hemodialysis patients. Likewise, *Mori et al. (2019)* reported
**Table 2  Correlation of risk factors with sarcopenia.**

| | Age (P-value) | BMI (P-value) | HD (P-value) | Speed (P-value) | Strength (P-value) | SMI (P-value) | Dep (P-value) | Cog (P-value) | PA (P-value) | Sarcopenia[a] (P-value) |
|---|---|---|---|---|---|---|---|---|---|---|
| Age | | .059 (.550) | .187 (.057) | −0.572 (<.001) | −0.520 (<.001) | −0.308 (.001) | 0.138 (.168) | −0.363 (<.001) | −0.265 (.006) | 0.458 (<.001) |
| BMI | 0.059 (.550) | | −0.078 (.430) | −0.134 (.174) | 0.096 (.330) | 0.644 (<.001) | −0.245 (.012) | 0.187 (.057) | 0.062 (.533) | −0.405 (<.001) |
| HD | 0.187 (.550) | −0.078 (.430) | | −0.166 (.092) | −0.126 (.202) | −0.136 (.169) | 0.180 (.068) | −0.194 (.049) | −0.183 (.062) | 0.220 (.025) |
| Speed | −0.572 (<.001) | −0.134 (.174) | −0.166 (.092) | | 0.583 (<.001) | 0.232 (.018) | −0.234 (.017) | 0.414 (<.001) | 0.204 (.037) | −0.460 (<.001) |
| Strength | −0.520 (<.001) | 0.96 (.330) | −0.126 (.202) | 0.583 (<.001) | | 0.540 (<.001) | −0.230 (.019) | 0.358 (<.001) | 0.102 (.301) | −0.503 (<.001) |
| SMI | −0.308 (.001) | 0.644 (<.001) | −0.136 (.169) | 0.232 (.018) | 0.540 (<.001) | | −0.167 (.089) | 0.405 (<.001) | 0.150 (.128) | −0.561 (<.001) |
| Dep | 0.138 (.163) | −0.245 (.012) | 0.180 (.068) | −0.234 (.017) | −0.230 (.019) | −0.167 (.089) | | −0.242 (.013) | −0.234 (.017) | 0.316 (.001) |
| PA | −0.265 (.006) | .062 (.533) | −0.183 (.062) | 0.204 (.037) | 0.102 (.301) | 0.150 (.128) | −0.234 (.017) | 0.086 (.386) | | −0.320 (.001) |
| Sarcopenia | 0.458 (<.001) | −0.405 (<.001) | 0.220 (.025) | −0.460 (<.001) | −0.503 (<.001) | −0.561 (<.001) | 0.316 (.001) | −0.329 (.001) | −0.320 (.001) | |

**Notes.**
[a]Sarcopenia: 0 defined as no sarcopenia, 1 defined as sarcopenia.

BMI, body mass index; SMI, skeletal muscle index; HD, duration of hemodialysis; Dep, depression; Cog, cognitive performance; PA, physical activity.

**Table 3  Results of logistic regression analysis of risk factors for sarcopenia.**

| Risk factors | OR (95% CI) | P-value | OR (95% CI) | P-value |
|---|---|---|---|---|
| Age | Reference <60 years | | | |
| Age ≥60 years | 0.193 (0.076–0.488) | .001 | | |
| Sex | Reference female | | | |
| Male | 0.941 (0.414–2.138) | .885 | | |
| Comorbidities | Reference no history of diabetes mellitus | | | |
| Diabetes mellitus | 2.812 (1.202–6.580) | .017 | 0.483 (0.195–1.198) | .116 |
| Duration of HD | Reference duration <5 years | | | |
| Duration 5–10 | 1.614 (0.553–4.713) | .381 | 1.613 (0.522–4.981) | .406 |
| Duration ≥10 years | 0.316 (0.111–0.896) | .030 | 0.366 (0.120–1.113) | .077 |
| Depression | Reference normal range (0–13 scores) | | | |
| Depression >14 scores | 3.229 (1.139–9.157) | .028 | 0.281 (0.090–0.880) | .029 |
| Physical activity | Reference moderate to high physical activity | | | |
| Low physical activity | 4.921 (2.044–11.850) | <.001 | 0.277 (0.110–0.699) | .007 |

**Notes.**
[#]Adjusted by age.

HD, hemodialysis.

that the DM was an independent factor for sarcopenia in Japanese patients undergoing hemodialysis (OR = 3.11). Association of DM and sarcopenia has been reported in several studies (*Kim et al., 2010*; *Park et al., 2006*). Insulin deficiency might be an important contributor to the development of sarcopenia (*Rasmussen et al., 2006*; *Wang et al., 2016*).

*Mori et al. (2019)* reported the prevalence of 40% and patients with DM shown a higher risk for sarcopenia than those without history of DM. It is also known that insulin deficiency leads to a reduction in protein synthesis in muscle mass (*Mori et al., 2019*). In addition, uremic sarcopenia has been proposed as skeletal complication, e.g., muscle wasting, in CRF (*Fujita et al., 2006*). Mitochondrial dysfunction in skeletal muscle in patients with CRF play an important role in promoting the uremic sarcopenia (*Nishi et al., 2020*). The induced mitochondrial damage can promote inflammation or cause the release of cytochrome; resulting in apoptosis and reduced ATP synthesis leads to cell damage. Therefore, these cellular processes may develop uremic sarcopenia in CRF patients (*Takemura, Nishi & Inagi, 2020*).

The present study found that sarcopenic patients had a lower physical activity compared to non-sarcopenic patients. It is known that loss of muscle mass occurs with ageing and causes primary sarcopenia. However, secondary sarcopenia has been described to be not only due to ageing but also to low physical activity, disease and nutritional factors (*Sabatino et al., 2020*). Regarding physical inactivity, it is possible to identify an individual who had a sedentary lifestyle, bed rest, for example. In addition, disease conditions such as organ failure disease, inflammatory disease have been observed in secondary sarcopenia. Increased protein degradation such as muscle wasting from CRF leads to protein energy wasting and contributes to malnutrition in patients with CRF. Moreover, decreased protein synthesis has been found in persons with low physical activity which results in muscle disuse and may contribute to sarcopenia (*Sabatino et al., 2020*). Therefore, both, high protein degradation and low protein synthesis may play a role in the development of sarcopenia in patients with CRF.

Besides, it is unknown whether decreased physical activity causes loss of muscle or muscle loss causes decreased physical activity. Additionally, sarcopenic hemodialysis patients had poor physical performance and/or muscle strength which again might increase physical inactivity and also cause depressive symptoms. *Vettoretti et al. (2019)* reported a higher prevalence of depression was associated with sarcopenia in CRF patients. It was suggested that these relationships might be explained by physical inactivity and malnutrition. Therefore, reduced physical activity causes depression and vice versa depression causes physical inactivity. In the present study it was found that physical activity was inversely associated with depressive symptoms, i.e., individuals with symptoms of depression had lower physical activity. Further studies need to explore the relationships between depression and physical activity in sarcopenic CRF patients.

Further studies are required to explore potential causative factors and mechanisms for development of sarcopenia in CRF such as inflammatory markers. Besides, an assessment of sarcopenia should be considered in individuals at increased risk for sarcopenia in hemodialysis. Secondary sarcopenia is associated with chronic illness states such as CRF and the present study supports that less physical activity can be observed in sarcopenic hemodialysis patients. It is known that low physical activity is a behavioral risk factor, therefore, the implementation of physical activity should be explored to prevent/treat sarcopenia.

The main limitation of the study was its cross-sectional design, use of a single center and relatively small sample size. Thus, a causal relationship of physical activity and depression with sarcopenia might be difficult to establish. Therefore, the prevalence of sarcopenia in hemodialysis patients from the study may have limited generalizability. Despite the limitation noted, the study extends pervious research that has been completed in sarcopenia among patients receiving hemodialysis and explore the relationships with physical activity and depression.

## CONCLUSIONS

Decreased physical activity might lead to symptom of depression and vice versa depression might contribute be physical inactivity and these also play an important role in sarcopenia among hemodialysis patients.

## ACKNOWLEDGEMENTS

The authors would like to thank all staff from the Hemodialysis Center at Thammasat University Hospital for their kind support. In addition, we would also like to thank the participants and their caregivers for participating in the study. We would like to pass on our thanks and appreciation to Prof. Nantika Thavichachart, M.D., Department of Psychiatry, Faculty of Medicine, Chulalongkorn University, Thailand for her permission to use the Thai version of the Beck Depression Inventory.

### Funding

This study was supported by Thammasat University Research Fund, Contract No. TUFT 016/2563. The funders had no role in study design, data collection and analysis, decision to publish, or preparation of the manuscript.

### Grant Disclosures

The following grant information was disclosed by the authors:
Thammasat University Research Fund: Contract No. TUFT 016/2563.

### Competing Interests

The authors declare there are no competing interests.

### Author Contributions

- Kornanong Yuenyongchaiwat conceived and designed the experiments, performed the experiments, analyzed the data, prepared figures and/or tables, authored or reviewed drafts of the paper, and approved the final draft.
- Sasikan Jongritthiporn and Kasarn Somsamarn conceived and designed the experiments, performed the experiments, analyzed the data, prepared figures and/or tables, and approved the final draft.

- Oranat Sukkho performed the experiments, prepared figures and/or tables, and approved the final draft.
- Sasipim Pairojkittrakul and Opas Traitanon conceived and designed the experiments, prepared figures and/or tables, and approved the final draft.

## Human Ethics

The following information was supplied relating to ethical approvals (i.e., approving body and any reference numbers):

The Research Ethics Committee on Human Subject of Thammasat University and the Ethics in Human Research Committee of the Thammasat University Hospital (COA No. 117/2562).

## Clinical Trial Ethics

The following information was supplied relating to ethical approvals (i.e., approving body and any reference numbers):

The Thai Clinical Trials Registry was approved.The Clinical Trials Registry (TCTR) is TCTR20190911004

## Ethics

The following information was supplied relating to ethical approvals (i.e., approving body and any reference numbers):

The Research Ethics Committee on Human Subject of Thammasat University and the Ethics in Human Research Committee of the Thammasat University Hospital (COA No. 117/2562).

## Data Availability

The raw measurements are available in the Supplemental File.

## Clinical Trial Registration

The following information was supplied regarding Clinical Trial registration:

20190911004

## Supplemental Information

Supplemental information for this article can be found online at http://dx.doi.org/10.7717/peerj.11695#supplemental-information.

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
