# Peer review of "Depression and low physical activity are related to sarcopenia in hemodialysis: a single-center study"

_PeerJ, doi:10.7717/peerj.11695_

## Round 0.1 · original submission · Minor Revisions

The two reviewers and I are most impressed with many aspects of your submission. However, they have highlighted a number of ways in which this manuscript can be further improved.

Reviewer 1 ·

Basic reporting

The manuscript was generally clear. Relevant background information was provided.

Title: The title may need to be revised, as the data is based on only 34/104 participants.

Abstract: Reflects and reports what the study is about.

Introduction: Relevant background information is provided. The study aim is clearly stated.
Edits
1. On the Introduction page, at the end of the Introduction paragraph, lines 67-68 need to be reworded as the following: “Therefore, the purpose of this study was to explore……”

Experimental design

The study fits within the aims and scope for the journal. The methods is well described.

Methods:
The Methods are described with sufficient detail and information.
Edits
1. The authors are not blinded to the reviewers. On the Introduction page (no page number provided), in section title Materials and Methods, lines 73, please state the full name of the Ethics Committee that provided ethical approval.

Validity of the findings

Conclusions are stated and linked to the research question.

Discussion:
The main results of the study are discussed and linked to previous research.
Edits:
1. Please discuss the strengths and limitations of your study

·

Basic reporting

The current manuscript aims to describe the prevalence of sarcopenia in 104 patients on hemodialysis.

However, there are some inconsistencies and I have some questions and specific comments provided below.

In regard to the basic reporting, I would recommend double-checking the manuscript for missing references, as a number of references are missing from both the introduction as well as the discussion (see also specific comments).

Experimental design

The outcomes of the manuscript are beneficial for the existing body of literature around sarcopenia and especially within patients on hemodialysis.

Validity of the findings

No comment

Additional comments

Specific comments:
Introduction:
- Please be consistent in the use of abbreviations (i.e. physical activity is abbreviated in the abstract as PA, but not in the body of the manuscript).
- There are multiple references missing in the introduction (i.e. line 53, line 55, line 59, and line 65). Please double-check the manuscript for missing references.
- Line 49: The second sentence in the introduction is somewhat confusing. The authors describe the prevalence of CRF stages 4 and 5 but then describe they are increased 'by'. Should this be increased to?
- Line 51: I would recommend rewriting “also, over 0.1 million “ to “Also, over 100.000 …”
Methods:
- Line 81: It is unclear why the authors have exclusion criteria for blood pressure and heart rate. It would be good to add the reasoning for this in the materials/methods section.
- Line 102: It is unclear how the SMM and SMI were measured. Was this done by BIA or DXA? And it would be beneficial if manufacturer etc. are specified.
Results:
- Line 129: The authors describe “The correlation of risk factors with mortality of sarcopenia”. I’m unsure where the mortality of sarcopenia derives from. This manuscript does not seem to measure mortality.
Discussion:
- Line 151: Please refrain from using the word elderly, but use older adults instead.
- Line 154: The authors describe that advanced age, long duration of hemodialysis, etc. are associated with sarcopenia and adverse health outcomes. It is unclear what the suggested adverse health outcomes are. These are never described, nor can I find these suggested results.
- Line 156-165: The authors describe the current body of literature around the prevalence of sarcopenia in hemodialysis patients. This would fit as-is, better in the introduction. In the discussion, it would be valued to relate this information with the outcomes of the presented research.
- There are multiple references missing in the discussion (i.e. line 168, line 184, line 194, line 196, line 204, and line 207). Please double-check the manuscript for missing references.
- Line 168: Please add a reference to this statement.
- Line 177: This sentence is a bit unclear and a rewording would be advised to ensure readability.

---

## Round 0.2 · Minor Revisions

While both the reviewers are happy to recommend acceptance of this manuscript, both Reviewer 2 and I feel there are a range of typographical errors that require correction for this can be accepted. Please look to upload a clean version of the manuscript as it appears that your track changes version is a number of errors with respect to additional line spaces being present. Further please correct the following small typographical errors in the track changes MS Word version:

Line 87: please remove the word "crisis".

Line 111 – 114: please reword this to ".... Omron Healthcare Co., Ltd., Japan), with body weight, BMI and SSM also recorded".

Line 233: change this to "The main limitation the study was its cross-sectional design, use of a single centre and relatively small sample size ".

Line 234-245: remove the following sentence "Further, single centre site study a relatively small sample size were enrolled".

Reviewer 1 ·

Basic reporting

The revised manuscript meets the criteria for this section.

Experimental design

The revised manuscript meets the criteria for this section.

Validity of the findings

The revised manuscript meets the criteria for this section.

Additional comments

Thank you for making the suggested revisions, I am satisfied with the revised manuscript.

·

Basic reporting

Basic reporting is good. The manuscript would benefit from an editorial check before publishing, otherwise good.

Experimental design

No comment

Validity of the findings

No comment

Additional comments

Thank you for responding to the comments. The revisions that were made are well explained and sufficient.

---

## Round 0.3 · accepted · Accept

I would like to thank the authors for their commitment to take on board the constructive criticisms of the two reviewers and I. I believe this manuscript is now of the standard that it can be accepted for publication in PeerJ.